# CADUCEO: A Platform to Support Federated Healthcare Facilities through Artificial Intelligence

**DOI:** 10.3390/healthcare11152199

**Published:** 2023-08-04

**Authors:** Danilo Menegatti, Alessandro Giuseppi, Francesco Delli Priscoli, Antonio Pietrabissa, Alessandro Di Giorgio, Federico Baldisseri, Mattia Mattioni, Salvatore Monaco, Leonardo Lanari, Martina Panfili, Vincenzo Suraci

**Affiliations:** Department of Computer, Control and Management Engineering “Antonio Ruberti”, Sapienza University of Rome, Via Ariosto 25, 00185 Rome, Italy; menegatti@diag.uniroma1.it (D.M.); giuseppi@diag.uniroma1.it (A.G.); dellipriscoli@diag.uniroma1.it (F.D.P.); pietrabissa@diag.uniroma1.it (A.P.); digiorgio@diag.uniroma1.it (A.D.G.); mattioni@diag.uniroma1.it (M.M.); monaco@diag.uniroma1.it (S.M.); lanari@diag.uniroma1.it (L.L.); panfili@diag.uniroma1.it (M.P.); suraci@diag.uniroma1.it (V.S.)

**Keywords:** deep learning, artificial intelligence, e-health

## Abstract

Data-driven algorithms have proven to be effective for a variety of medical tasks, including disease categorization and prediction, personalized medicine design, and imaging diagnostics. Although their performance is frequently on par with that of clinicians, their widespread use is constrained by a number of obstacles, including the requirement for high-quality data that are typical of the population, the difficulty of explaining how they operate, and ethical and regulatory concerns. The use of data augmentation and synthetic data generation methodologies, such as federated learning and explainable artificial intelligence ones, could provide a viable solution to the current issues, facilitating the widespread application of artificial intelligence algorithms in the clinical application domain and reducing the time needed for prevention, diagnosis, and prognosis by up to 70%. To this end, a novel AI-based functional framework is conceived and presented in this paper.

## 1. Introduction

Deep learning (DL) is a branch of artificial intelligence (AI) that relies on artificial neural networks (ANN) capable of learning from large amounts of data and extracting information relevant to various tasks. DL has found numerous applications in the medical field, particularly in imaging diagnosis [1], clinical and drug research [2], disease classification and prediction [3], personalized therapy design [4], and public health monitoring [5]. It can offer significant advantages over traditional data analysis methods, both in terms of performances and automation [6]. Moreover, it can also complement and enrich the knowledge and skills of healthcare professionals, providing them with evidence-based clinical decision support tools. However, DL also presents challenges and limitations related to the following:The need for quality and representative data;The difficulty of interpreting and explaining the results given as output by ANNs, i.e., explaining their inner workings;The legal and ethical implications arising from the use of intelligent systems in clinical practice.

Having domain-specific representative data has always been the bottleneck for data-driven methods; if ANNs are trained over data which are not statistically informative with respect to the problem being addressed, then, even if they exhibit pleasing behavior over training data, they may fail to generalize when applied in a real-world scenario.

This problem, although common, becomes crucial in the medical field where most clinical data are protected by privacy laws, such as the General Data Protection Regulation (GDPR) [7], and cannot be used without the patient’s consent. Furthermore, even if the patients give consent, their data need to be anonymized before being used to train ANNs, so that it is not possible in any way to track it back to the single patient. Additionally, even if data are maintained by hospitals or other clinical institutes and have previously been anonymized, ethical approval must be obtained before using it.

To this end, federated learning (FL) [8] offers a valid strategy to overcome most of the above-mentioned problems. FL is a solution for distributed machine learning (ML) problems which allows a federation of clients sharing the same machine learning model to learn a model without any exchange of data, in a privacy-preserving and GDPR-compliant way. It can allow exploitation of the vast amount and variety of medical data available in different sources, increasing the statistical power and generalizability of ML models while also providing a solution to overcome privacy, security, and data governance issues that prevent the sharing of medical data between various healthcare institutions. FL has been successfully applied to various clinical fields, such as imaging diagnosis, drug research, and genomics.

Although FL makes it possible to overcome data sharing issues, and to implement strategies such as Continuous Learning (CL) [9], the lack of explainability of the inner workings of ML models, such as ANNs, is a major limitation. Explainable AI (XAI) solutions [10] offer valid tools to explain the functioning of ML algorithms in a way that humans can easily understand and interpret. Most of the time, the specific XAI solution depends on the particular type of data being processed, such as Grad-CAM [11] for convolutional neural networks (CNNs) for image classification; however, data type-independent solutions, such as LIME [12] or NAMs [13], can also be considered.

Making the inner workings of ML models interpretable to humans is the first step toward Trustworthy AI which focuses on creating AI systems which are reliable and respect human rights. To this end, XAI can help ensure that AI systems are robust, reliable, and ethically sound.

This work proposes a novel functional architecture for the introduction and use of AI algorithms in clinical practice. In particular, we will focus on the project “Cloud plAtform for intelligent prevention and Diagnosis sUpported by artifiCial intelligEnce solutiOns” (CADUCEO) which aims to design and develop a cloud platform for the realization of medical applications for the prevention, diagnosis, and prognosis of diseases, integrating advanced AI technologies and solutions with reference to three diseases of the digestive system: eosinophilic esophagitis [14], chronic inflammatory bowel diseases [15], and portal hypertension [16].

The main contribution of this paper is in the discussion of the functional architecture designed in the project CADUCEO. The project proposes a federated platform which runs based on FL algorithms, which allows the cohort medical centers to foster the performance of their machine learning-based algorithms by sharing the knowledge they locally generate, without having to share privacy-sensitive medical information. In addition, this paper introduces the main machine learning-based algorithms developed in the CADUCEO project for providing decision support to medical operators in some operations, including automated image processing (e.g., for image inpainting and cell counting), data augmentation for improving the classifiers’ detection accuracy, and support to diagnosis through automated image classification. In conclusion, this paper provides a broad overview of the role that the latest AI algorithms can play in improving the efficiency of medical operations, in terms of agility and improved quality of treatment.

The remainder of this paper is organized as follows. Section 2 reports used materials and methods. Section 3 details the results of the system design. Section 4 discusses the functionalities and the considered use cases. Finally, Section 5 draws the conclusion and future works.

## 2. Materials and Methods

### 2.1. Data Augmentation

In the machine learning literature, data augmentation techniques are commonly classified into two main categories [17]. The first category involves methodologies that directly manipulate images, such as algebraic–geometric transformations [18], color modifications [19], noise addition [20], image combinations [21], and random deletions [22]. However, these methods are known to have some drawbacks and often result in marginal performance improvements that may not justify their usage [23].

On the other hand, the second category employs deep learning techniques for data augmentation, which includes increasing the feature space [24], adversarial training [25], and generative adversarial networks (GANs). GANs have gained significant popularity in the clinical field since their introduction in 2014 [26]. These networks have the ability to generate new training data, leading to substantial performance enhancements for classifiers compared to training on the original, limited dataset. GANs have been successfully applied to various medical applications, including the study of electroencephalograms [27], brain tumor detection [28], and analysis of dermatological lesions [29]. Moreover, diversified GAN architectures have been utilized in [30] for risk stratification and disease classification in rare clinical cases, namely, primary Sjögren’s syndrome and hypertrophic cardiomyopathy. Meanwhile, traditional algebraic transformation methods continue to find applications in the medical field, such as in [31], where five different direct image manipulation methods are analyzed for prostate cancer detection.

While there are not many studies systematically comparing various data augmentation techniques, a study [32] demonstrates that the most effective methods include geometric transformations (rotations, flips, and cropping) and GANs.

In recent years, a promising class of methodologies known as meta-learning has gained popularity [33]. These methods rely on the principle of neural architecture search [34], wherein neural networks optimize other neural networks. Specifically, optimization mechanisms are employed to tune the hyperparameters of a neural network, maximizing classification accuracy. Meta-learning and neural network optimization for data augmentation are addressed in [35,36], where a double-cascade neural network architecture is proposed, with the second network trained to optimize the parameters of the first one.

### 2.2. Federated Learning

Federated learning (FL), which is distributed and protects privacy, is a great option for healthcare applications. FL has a wide range of applications in situations where cooperation among stakeholders, including hospitals, laboratories, and governmental organizations, would considerably enhance the management of particular issues. The management of electronic health records (EHRs), remote health monitoring, and medical imaging are a few examples of such applications [37].

EHRs digitally archive patient health data, such as diagnosis, treatments, and analyses [38]. They serve as an important source of information that could help with disease diagnosis and evaluation. Sharing EHRs among various institutions can, however, provide regulatory difficulties. The authors of [39] suggest a collaborative learning protocol (PRCL) based on the FL framework that is privacy aware and resource efficient.

In order to avoid data breaches, PRCL employs a model-splitting technique in which the most computationally intensive portion of the learning process is delegated to a cloud server.

The FL-based method SplitNN is introduced in [40] which also presents a model-splitting strategy. The plan is to divide the model into various components, each of which will be trained by a different customer, avoiding a direct data exchange. Reference [41] uses data from patient EHRs to predict hospitalizations for cardiac events using an FL setup based on the soft-margin l1-regularized sparse Support Vector Machine (sSVM) classifier and an iterative cluster Primal–Dual Splitting (sPDS) method.

In order to solve privacy concerns, ref. [42] introduces an FL framework built on a differential privacy mechanism that has been successfully used with real-world EHRs from one million patients. In [43], where synthetic sounds are applied to the parameters at the clients’ side prior to aggregation, differential privacy is also used. Federated-autonomous deep learning (FADL), an FL technique, is utilized in [44] to forecast patients’ death using their EHRs. The innovative feature is that some parts of the model are trained using distributed distribution of data from all sources, while other parts are trained using specific sources.

With the introduction of the stochastic FL technique known as stochastic channel-based federated learning (SCBF), the prediction of patients’ mortality as a binary classification problem is investigated in [45]. By stochastically choosing the clients with the biggest local gradients, it is possible to update the server model’s weight while maintaining privacy.

### 2.3. Explainable AI

Explainable artificial intelligence (XAI) is a set of methods and processes that allow users to understand and trust the results produced by machine learning algorithms [46]. Generally, the inner workings of AI algorithms are inaccessible and/or incomprehensible to humans. However, in certain applications, it is necessary to provide clear explanations for the reasons behind a specific output. Moreover, the interpretability of an algorithm’s output enables users to provide feedback that can be used to improve its accuracy and performance [47].

The main limitation of XAI lies in the empirical trade-off relationship between prediction accuracy and interpretability: as the model becomes more performant, it becomes less interpretable [48].

There are three categories of XAI methods [49]:Intrinsically Interpretable Methods: These methods use inherently interpretable models, such as Decision Trees and Linear Regression.Sample-Based Methods: Specific samples from the dataset are used to describe the model’s behavior. An example is the K-Nearest Neighbors method.Model-Agnostic Methods: The explanation method is separate from the model being explained. Examples include Global Surrogate and Local Surrogate.

Methods in the first category generally achieve lower performance because, as mentioned earlier, high interpretability is traded-off with prediction accuracy. Possible solutions to this problem include using specific sample sets to describe the model (Category 2) or decoupling the explanation method from the model (Category 3), which will be discussed further.

Perceptive Interpretability methods generate elements assumed to be immediately interpretable [50]. For example, the Saliency Method explains the algorithm’s decision by assigning values that reflect the importance of input components in contributing to the decision. This can be achieved through probabilities or heatmaps [47].

Signal Methods monitor the activation of specific neurons or neuron groups, allowing the activated values to be manipulated or transformed into interpretable forms. For example, the activation of neurons in a given layer can be used to reconstruct an image similar to the input. Signal Methods are considered interpretable as they connect the generation of a certain output to a reduced set of influential features. Deconvolution is often used to make the output more interpretable [50].

Verbal Interpretability methods aim to provide fragments of text that are naturally understandable to humans. These methods seek to create sentences that indicate logical causality relationships, like “If A happens, then B happens”, or suggest actions based on certain conditions, like “If condition A occurs, then execute action B”. The output is represented as a binary sequence, which follows a probability distribution [50].

Interpretability via Mathematical Structure methods generate outputs that require additional cognitive processing by humans compared to the immediately interpretable outputs of Perceptive Interpretability methods.

Predefined Models allow the study of systems with unknown behavior by defining parametric models with relevant and easily interpretable terms. The simplest interpretable predefined model is the linear combination of variables. The Generalized Additive Model (GAM) is a generalization of linear models that includes easily interpretable terms [50].

## 3. Results

In light of the previous discussion, in this section, we will detail the overall functional architecture that we propose for the design of a software platform able to support healthcare operators in their decisions.

To cope with the data paucity, the complexity of the ML model training and to ensure GDPR compliance and trustworthiness of the overall system, we have identified five key functionalities, gathered in the so-called AI Layer, whose logical interaction is depicted in Figure 1.
The data pre-processing system is a functional block that contains all the software solutions for data and clinical image enhancement. These solutions may employ both classical computer vision techniques, such as de-noising [51] and feature selection [52], and deep learning techniques, such as deep image inpainting [53], depending on the nature and characteristics of the data being processed. Data transformation and curation is one of the fundamental steps behind the design and implementation of any data-driven system and has to be designed in such a way that the integrity and correctness of the data to be analyzed are preserved.The synthetic data generator block gathers the functionalities related to the generation of synthetic, or artificial, data, whose function is to support and improve the training process of the AI functionalities. The role of synthetic data is of central importance in all applications where the availability (and quality) of real data is limited, as is common in the medical domain. Starting from publicly available data, as well as anonymized data samples coming from clinical facilities and deep learning solutions, including generative adversarial networks [54] and deep reinforcement learning [55], appropriately specialized on the specific clinical use cases, synthetic data will be generated. And it will undergo a review process by clinical experts to assess its quality to increase the trustworthiness of the overall trained system.Federated learning (FL) has been identified as an enabling technology for the distributed and decentralized training of the ML models underlying the platform’s AI capabilities. As the solutions of the CADUCEO framework are developed for clinical use cases, it is crucial to ensure their full compatibility with the GDPR and to observe in their design all appropriate precautions related to the protection of patient privacy. In this respect, FL makes it possible to ensure collaboration between different institutions, including prospective customers interested in improving certain functionalities, without the need for any exchange of clinical data. Integrating an FL system into the AI Layer thus ensures its maximum compatibility in terms of adoption and further developments.The explainable AI framework and methods functional block contains all the software features and guidelines to direct the design and development of AI solutions from the perspective of the explainability and interpretability of their results. Indeed, it is critical to the goals of the CADUCEO framework to offer clinical experts not only a set of recommendations and suggestions but to supplement them with adequate information critical to their correct interpretation.The training of machine learning models block, representing from a functional point of view the whole design process of the machine learning models (such as deep learning solutions), deals with the process of training them on the basis of available data (from project sources, public sources, and synthetic ones). The training of the models will take place under the constant supervision of clinical experts so that aspects related to the interpretability of the results and the quality of the proposed suggestions can be improved.

The figure details how these functional blocks are interconnected, the main functional interfaces with the Platform Layer (mainly related to the fruition of data by machine learning models) and to the APP Layer (mainly related to the interface of AI Layer functionalities with the clinical staff).

The following sections provide a survey that covers the main results and techniques available to reduce the amount of data required for the ML training. Such solutions involve data pre-processing, augmentation, and federated learning.

## 4. Discussion

### 4.1. Data Pre-Processing

In clinical settings, the first major problem faced when performing statistics or using AI systems is that of managing large amounts of data. The latter are often raw, heterogeneous, and come from different hardware technologies, aspects that make it difficult for a human operator to process them manually. The need to process huge amounts of data (Big Data) has quickly led to the development of a specific branch of mathematics and computer science known as data mining [56], whose main focus is to extract patterns from large and unstructured datasets. Thus, an attempt is made to bring order to the Big Data environment to create ordered datasets ready to be exploited for traditional statistical analysis or supervised and unsupervised learning algorithms [57]. The performance and quality of knowledge extracted by a data mining method in any framework depends not only on the method itself but also on the nature of the data itself. Factors such as noise, missing values, inconsistent and redundant data, and duplicates and multi-repeats strongly affect the performance of the algorithms used to learn and extract knowledge. Indeed, it has been widely demonstrated that low-quality data lead to low-quality knowledge and poor performance regarding artificial intelligence systems [58]. Thus, the importance of data pre-processing seems evident: this is a preliminary stage prior to data mining, the primary goal of which is to obtain final datasets that can be considered correct and useful for data mining algorithms and, subsequently, ML [57]. When few datasets are available, it is simple and easy to manually pre-process them, eliminating inconsistencies and possible measurement and/or transcription errors. Pre-processing Big Data constitutes a challenging task, as manual clean-up is not possible. Therefore, it is necessary to design data pre-processing strategies that can appropriately and sequentially filter out incorrect data.

Various data pre-processing techniques can be used to address a multitude of raw data issues, such as the ones listed below:Missing data. It is often the case that, during data acquisition, some features could not be recorded for several records. In this case, we speak of missing data, which therefore cannot be used as is for data mining. There are several approaches to deal with this problem, first of all eliminating all instances that are stumped [59]. However, this method results in a substantial thinning of the dataset itself, resulting in worse performance when (for example) training a neural network. Other more refined methods, such as the ones employed in this work, make use of statistical procedures such as maximum likelihood to model each feature, as if it were a random variable, and then include an estimate of the missing numerical values. In this case, the drawback lies in the “goodness-of-fit” of the random model, which may even deviate greatly from the true value of the stumped data [60].Noise in the data. Even when there are no missing data, those that are present are often characterized by imperfections and inaccuracies, mainly due to measurement errors committed by hardware instruments or, especially in the clinical setting, by human operators. In these cases, it is possible to intervene by constructing noise filtering or data polishing algorithms [61], with the purpose of automatic noise detection and the consequent choice of intervention, which may reside in the modification of the data or, in extreme cases, its elimination.Feature selection. In every dataset, each record is characterized by countless features, which often turn out to be useless for the purpose and/or redundant. Feature selection algorithms allow features with high correlation to be eliminated, an aspect that results in lowering the risk of overfitting and speeding up data mining processes [62].Record selection or generation. Shifting the discussion from features to records, high correlation can also occur between data [63]. In the medical domain, for example, it may be the case that two different patients have very similar medical records, and as a result, for some machine learning or data mining applications, knowledge of only one clinical status may suffice. Conversely, it may be the case that a data acquisition campaign has collected too few or too little variety, so that manual augmentation of the dataset itself is necessary. Such augmentation techniques will be discussed at length in the next section.Class imbalance. In many supervised learning applications whose goal is classification, there is a significant difference among the absolute number of records of different classes in a classification problem. This situation is known as class imbalance [64]. The presence of class imbalance leads to a net imbalance toward the majority class and, as a result, there is a higher rate of misclassification for all other classes, which are in the minority. In these cases, pre-processing techniques can be used to resample the data to balance the class distribution. Within resampling, two main groups can be distinguished [65]: (i) undersampling, through which majority instances are removed to create a balance, and (ii) oversampling, which creates a superset of the original dataset by replicating some instances or creating new instances from existing ones.The application of data pre-processing techniques in the medical context improves clinical interpretation, exempts healthcare personnel from manual data cleaning, reduces computational costs, and improves the performance of predictors [66]. The study conducted in [67] found that the use of pre-processing is most common in critical disciplines of medicine, such as cardiology, endocrinology, and oncology. Regardless of the relevant clinical field, the most commonly used techniques are data reduction due to noise and/or the absence of data, and feature selection.

### 4.2. Data Augmentation

Deep convolutional neural networks constitute a widely used methodology for image classification in a variety of domains, including the clinical domain [68]. However, these networks are highly dependent on the presence of large amounts of data, which are necessary to train the networks to correctly classify images in order to avoid the phenomenon of overfitting [69], which is the tendency of a neural network to fit the training data far too well, losing the generalization capabilities. In the medical field, very often, it is not possible to have access to a sufficient amount of data for the proper training of neural networks. In this respect, data augmentation practices may be crucial when dealing with clinical analysis [23,70].

In the literature, two main classes of methods for performing data augmentation are distinguished [17]. The first one belongs to methodologies based on direct image manipulation: in this family, we find methods such as algebraic–geometric transformations, color modifications, noise addition, image combination, and random deletions [17]. Although these methods are known to suffer from some drawbacks and often low performance gains that do not justify their use in some domains, traditional algebraic transformation methods also remain widespread in the medical field: in [31], the authors analyze five different methods belonging to the family of direct image manipulation for prostate cancer detection. The second class makes use of deep learning to perform data augmentation: examples include the feature space augmentation [24], adversarial training [25], and generative adversarial networks (GAN) [26] methods. The latter has been one of the most widely used approaches in clinical settings since its introduction in 2014. Such networks have the ability to generate new training data that allow, for the same classifier, significantly better classification performance than using the original dataset. The method of GANs has been applied to the study of electroencephalograms [27], the detection of brain tumors [28], and dermatological lesions [29]. GANs diverse in network structure are employed by the authors in [30] for risk stratification and disease classification in two rare clinical cases, namely, primary Sjögren’s syndrome and hypertrophic cardiomyopathy.

Although there are not many studies that perform a systematic comparison of the various data augmentation techniques, in [32] the authors show that the most effective methods are those based on geometric transformations (rotations, flips, and cutoffs) and GANs.

In recent years, very promising new methodologies, called meta-learning [33], are gaining popularity. These methods are based on the principle of neural architecture search [34], which uses neural networks to optimize other neural networks. Specifically, optimization mechanisms are employed to perform tuning of the hyperparameters of a neural network so as to maximize the accuracy of classifications. Data augmentation by meta-learning and optimization of neural networks are addressed in [35,36]. In these two papers, the use of a dual neural network cascade can be seen: the second is trained to optimize the parameters of the first.

In other cases, however, the optimization of network parameters is sometimes carried out by making use of a class of algorithms belonging to one of the three branches of machine learning, namely, reinforcement learning [71]. For example, AutoAugment [72] is a reinforcement learning algorithm that seeks an optimal data augmentation “policy” among a constrained set of geometric transformations with various levels of distortion. In reinforcement learning algorithms, a policy is analogous to the strategy of the learning algorithm. This policy determines what actions to take in given states to achieve a goal. The AutoAugment approach learns a policy that consists of many sub-policies, each consisting of a specific image transformation, namely, to associate a geometric transformation to the given data as input, i.e., the horizontal flip of an input image. Reinforcement learning is then used as a discrete search algorithm to perform data augmentation optimally. In another paper [73], the authors show how it is possible to improve the results of meta-learning by reinforcement learning by selecting transformations using, instead of the entire dataset, only small parts of it.

Meta-learning by reinforcement learning is a technique recently proposed in the literature and still little explored [74]. One of the major challenges concerns the design of meta-learning schemes dedicated to the specific application being considered: certain architectures may be valid for some clinical cases and prove unsuitable for others. Another limitation lies in the problem of gradient nulling [75], a very frequent phenomenon that prevents the training of the network. Future developments, which are of interest to the CADUCEO project within the scope of the case studies defined therein, lie in the development of deep reinforcement learning algorithms [76] for the purpose of both solving optimization problems aimed at parameter tuning and selecting the data augmentation method most suitable for the specific clinical case study.

### 4.3. Federated Learning

This section details the concept and architecture of federated learning schemes, emphasizing its importance in the medical field and the characteristics, in terms of GDPR compliance and Privacy Assurance, that make it an ideal tool in applications using clinical data—or data which are confidential in nature.

Federated learning (FL) constitutes a distributed machine learning solution employed for solving machine learning problems in the absence of a data exchange. First introduced in 2016 [59] as an alternative to conventional approaches for training deep models based on data from mobile devices, it was readily applied in the field of mobile keyboard prediction for GBoard [77].

The federated reference architecture for FL consists of a set of devices, also called clients, in communication with a server, that drives the entire learning process, which relies on local client data. In contrast to a typical distributed optimization scenario, where a central entity distributes data to peripheral ones and governs the training process, in the federated one there is no exchange of data, and training is based solely on the clients’ data “as it is” (Figure 2); because the latter are strictly dependent on the use made by the device owner, they will be as follows:Non-IID (not independent and identically distributed), i.e., not representative of the statistical properties of the population identified for the problem under consideration;Unbalanced, from the point of view of numerosity, because the users do not behave in the same way;Highly distributed, because the number of devices participating in the training process may be greater than the total number of data available;Communication-limited, because they may not be accessible during training.

Under the assumption that the server and the devices share the same neural network model (in terms of architecture), the FederatedAveraging (FedAvg) algorithm first chooses a subset of the clients that will participate in the training process, and then sends the model, with the given initial parameters, to the clients. The clients perform the training phase of the model based on their own local data—the training step—and send the resulting local model back to the server. The server updates its model by performing the weighted average of the parameters of the local models received from the clients—the averaging step—and send the resulting model back to the clients. By iterating the two described steps, it can be shown that the model converges to a neural network with similar performances to a neural network centrally trained on the whole dataset.

The distributed nature and privacy protection, strengthened by the absence of a data exchange, of FedAvg make FL an ideal choice for medical applications, given the limitations imposed by the GDPR [79]. To date, FL is employed not only for predictive purposes based on information in electronic patient records, predicting hospitalizations for cardiac events [41], patient mortality [44], and ICU length of stay [80], but also in clinical research activities, such as the investigation of brain structural relationships [78], analysis of brain tissue from MRI images [81], and activities supporting clinical diagnostics, such as detection of COVID-19 from CT scans [82,83] or chest X-ray [84].

In addition to FL solutions developed ad hoc for the particular application under consideration, such as the previous ones, theoretical solutions have been proposed in the last period, aimed at enhancing FedAvg. Among them, FedProx [85] allows devices with optimization characteristics strongly different from others to be included in the optimization process, AdaFed [86] replaces the averaging mechanism of clients with a dynamic and adaptive heuristic weighting mechanism based on client performance, FOCUS [87] replaces the averaging mechanism with a credibility mechanism based on Shapley values [88], DecFedAvg [89] and CFA-GE [90] represent the first attempts to decentralize FedAvg by allowing each client to communicate with its neighbors, and FedLCon [91], the first consensus-based FL algorithm.

As part of the CADUCEO project, the interconnection of the participating hospitals in a federated manner is being evaluated, in such a way so as to increase the amount of data available, in full compliance with the privacy of patients, and then with current regulations, as well as the statistical representativeness of the data considered in relation to the use cases under consideration.

### 4.4. Use Cases

This section describes the main use cases investigated for the design of the first iteration of the software tools developed in the scope of the CADUCEO project, highlighting their clinical features and sketching the ad hoc AI functionalities under development.

#### 4.4.1. Eosinophilic Esophagitis (EoE)

Eosinophilic esophagitis is a chronic inflammatory disease located in the esophagus, characterized by an elevated number of a particular type of white blood cells, called eosinophils, that cause its dysfunction [14]. It is one of the most frequent esophageal diseases in children and young adults, with an incidence of 20 new cases per 100,000 inhabitants per year.

The diagnosis and monitoring of this disease is based on the identification, by means of esophageal biopsies, of the concentration of eosinophils. The presence of eosinophils is detected by staining with haematoxylin-eosin.

Specifically, the diagnosis of EoE requires manual microscopic inspection of biopsies with a diagnostic threshold of at least 15 eosinophils in at least one high-magnification field. The protocol requires identifying the area of tissue with the greatest infiltration of eosinophils and quantifying the peak eosinophil count (PEC) to compare it with the mentioned diagnostic threshold, taking care to distinguish intact eosinophils, i.e., those with clearly visible cytoplasmic granules and nucleus, which contribute to the PEC, from non-intact eosinophils, which do not.

In addition, EoE monitoring is based on the inspection of endoscopic images looking for one or more macroscopic features, i.e., exudates, rings, edema, longitudinal sulci, stenosis, for example, in such a way so as to associate each of them with a value of the Endoscopic Reference Score (EREFS) [92].

Automatically counting the eosinophils together with staging the pathology on the basis of the EREFS score would obviously significantly speed up the diagnosis and treatment processes. In both cases, GAN-based deep image inpainting techniques, such as [93] as well as Conditional GANs (cGANs) [94], are employed for data augmentation purposes; in the first case, the automated count fits in the general object detection branch of computer vision, with streamlined solutions such as YOLO [95] offering great opportunities in terms of performance and ease of training, while the EREFS-based classification is carried out by means of deep reinforcement learning algorithms, such as [96], in addition with XAI solutions, such as Grad-CAM [11]. Finally, FL methodologies, such as [91], are employed to increase the numerosity of training data allowing for the distributed training of the federation of hospitals, as well as for continual learning, namely, to prevent performance degradation over time.

#### 4.4.2. Inflammatory Bowel Disease (IBD)

Chronic inflammatory intestinal diseases are characterized by chronic inflammation of the intestine, due to an improper interaction between the intestinal microbiota, i.e., the set of symbiotic microorganisms that coexist with the human being without damaging them, and the mucosal immune system, i.e., the set of organs and cells specialized in defending the organism from external agents localized at the level of the intestinal mucosa. Depending on their characteristics and the intestinal tract affected, they can be divided into [15] the following:Ulcerative Rectocolitis (UCR), characterized by an inflammatory process localized exclusively at the level of the colon, which starts in the rectum and may extend continuously to the cecum;Crohn’s disease (MC), characterized by inflammation that can affect the entire intestinal tract and involve all layers of the intestinal wall, leaving several healthy sections between the inflamed areas;Unclassified chronic inflammatory bowel disease, a term reserved for those cases in which anatomopathological features common to both UC and MC are present.

The main risks of such diseases are essentially twofold: on the one hand, we find that of resective surgery, i.e., the reasoned removal of a more or less long segment of the intestine, and, on the other hand, we find the development, after approximately ten years of disease, of dysplastic lesions (cancer) in the intestinal and colic mucosa.

To date, the only diagnostic tool is the evaluation of inflammation indices and the performance of instrumental examinations in a cyclic manner. Building a predictive model able to classify autonomously the staging of the disease in terms of the Geboes score [97], the Robarts histopathology index [98] on the basis of histopathology data, and in terms of the Mayo score [99], Ulcerative Colitis Endoscopic Index of Severity (UCEIS) [100], Harvey Bradshaw Index [101], and Simple Endoscopic Score for Crohn’s Disease (SES-CD) [102] on the basis of endoscopic data would represent a step forward in terms of speeding up treatment and prognosis processes. To this end, GAN-based deep image inpainting methodologies, such as [103], deep reinforcement learning for classification, and FL are employed.

#### 4.4.3. Portal Hypertension (PH)

Portal hypertension (PH) is a pathological condition resulting from the evolution of chronic liver disease that results in an excessive increase in portal venous pressure [16], i.e., an increase in the pressure gradient between the portal vein and the inferior vena cava above the nominal value of 1–5 mmHg.

This increase leads to the formation of new collateral venous circles, called esophageal varices, in the esophagus and stomach, with a consequent increased risk of hemorrhage. While in pediatric age the most frequent form is pre-hepatic, caused by portal vein thrombosis, in adulthood the most frequent form is intrahepatic, commonly called cirrhosis.

Building an ML model able to estimate PH values on the basis of instrumental data, thus making a prediction about the prognosis of the disease, would be beneficial in terms of diagnosis and treatment processes. To this end, deep regression models, such as [104], will be employed, together with FL techniques for training and performance keeping purposes.

## 5. Conclusions

This paper has introduced the e-health project CADUCEO “Cloud plAtform for intelligent prevention and Diagnosis sUpported by artifiCial intelligEnce solutiOns”, which aims to design and develop novel machine learning based tools for assisting medical operators in improving the efficiency of today’s medical practice. The paper has presented the functional architecture designed by the project partners, which is centered around the idea of the federation of cooperating medical centers which, through federated learning algorithms, can share knowledge and improve the respective processes, without having to share sensitive medical data. Four key AI-based algorithms are developed and made available through the CADUCEO platform. They include algorithms for support to diagnosis through the automated analysis of medical images and other data; algorithms for automated counting of cells (as a relevant example of the contribution that AI can give to speed up the current, very time consuming tasks of medical practice); algorithms for image processing, focusing on the removal of undesired artifacts that might be present in the image, to make them usable; and algorithms for data augmentation, which allow to expand artificially the amount of data available for training the AI-based algorithms.

Finally, the three use cases that are the focus of the CADUCEO project have been presented, discussing the impact that the above AI-based tools can bring.

## Figures and Tables

**Figure 1 healthcare-11-02199-f001:**
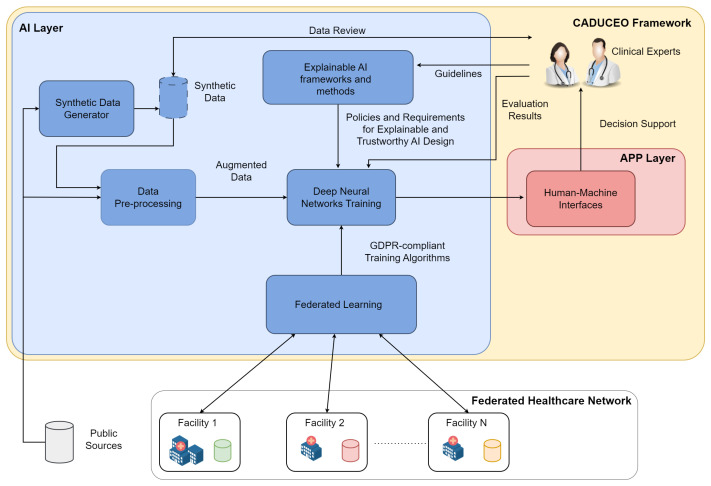
Generalarchitecture of the proposed distributed AI system.

**Figure 2 healthcare-11-02199-f002:**
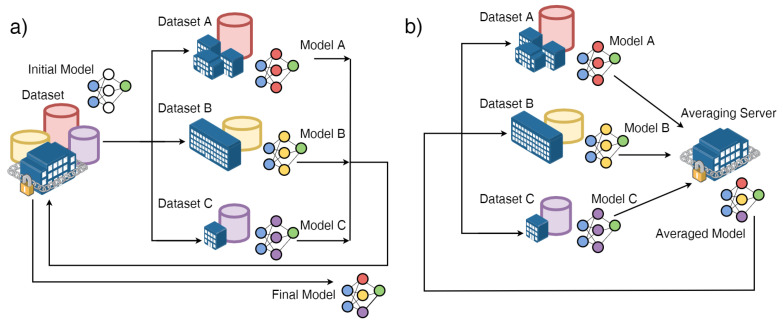
(**a**) Distributed optimization scenario, in which a centralized entity distributes data to peripheral entities and governs the training process; (**b**) federated optimization (FL) scenario, in which there is no data exchange. Image derived from [78].

## Data Availability

Not applicable.

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
