# Peer review of "CADUCEO: A Platform to Support Federated Healthcare Facilities through Artificial Intelligence"

_healthcare, 2023, doi:10.3390/healthcare11152199_

Round 1

Reviewer 1 Report

Dear Authors,

Thank you very much for your exciting contribution to the journal.

Below you will find some comments and suggestions that (I hope) will help you to improve the quality and soundness of your manuscript.

Introduction

Line 18: “...performances and automation” please include a reference to sustain this statement.

Line 24: “form the sue,” should it be “from the use”?

Line 33: “its” should be “their”.

Lines 34-36: The phrase that makes up these lines is difficult to follow. Please consider rephrasing it.

Line 37: “ [7] offer” should be “offers

Lines 41-44: This sentence is too long. Please consider rephrasing it. 

Line 50: “solutiong” should be “solution”

Line 64: Kindly add a reference for each disease. 

Line 67-68: Please review if it is needed to repeat the term “federated” so often.

Line 70: Consider removing “(by analyzing their data)” since it is already understandable what you mean. 

It would be interesting to see discussed in this section, if possible, some federated learning systems that are already implemented - in terms of their goals, results, and clinical acceptability.

Platform Design

Lines 91-98: Please consider including a reference for each pre-processing technique introduced here. 

Line 104: “... Deep Reinforcement Learning”, please include a reference. 

Line 106-108: “The data thus generated will then be reviewed by clinical experts”... will it? Because in Figure 1, the clinical experts seem only to receive information regarding decision support. Please clarify.

Data Pre-Processing

Lines 145-146: “The main goal of all (...) that appear chaotic and uncorrelated”. This phrase gives the idea that the goal is to extract patterns that are like that. That is not the case. Even though the patterns often have those characteristics, it is not a goal that they are like that. 

Lines 157-158:  “... machine learning” - since the acronym “ML” was already introduced, use it. 

Line 189: “...the case that two different patients have very similar….”, I am sorry, but I don’t think I’m being able to follow the rationale behind this example. Are the authors suggesting that removing the records from one of the patients might improve the algorithm’s performance? Even though they are different people? I kindly suggest a clarification.

Lines 197-198: “...that is, among the prob…”, please consider removing this part of the phrase in order to improve readability.

Data Augmentation

This is a very interesting and well-written section. My two main comments are:

1. a suggestion of changing lines 238-240 to right before introducing GANs (after discussing geometric transformations)

2. Clarify the interesting and important discussion done in lines 252-261. Maybe by giving some actual examples of what a “policy” might be.

Federated Learning

Line 289: “Figure 3”, should be “figure 2”.

Line 292: “identiied”, should be identified.

Line 308: “sugared”, I suggest the use of another term. 

Line 309: “Federated Learning”, please use the already defined acronym.

Line 311: “think of”, I think this expression can be removed.

Use cases

Three different diseases (cases) are presented. However, the impact of FL is only understandable for the first one (speeding up the counting of cells). For the remaining two “use cases”, it is not clear what FL has to offer or how it is being applied. Since this is a key point of the paper, I suggest that the authors deeply describe how FL is being applied to each of the three “use cases”.

No comments.

Reviewer 2 Report

This work proposes a functional architecture for the introduction and use of AI algorithms in medical practice. The focus was on the project "Cloud platform for intelligent prevention and Diagnosis sUpported by artifiCial intelligEnce solutiOns" (CADUCEO), which aims to develop a cloud platform for the prevention, diagnosis, and prognosis of diseases, integrating advanced AI technologies and solutions with reference to three diseases of the digestive system: eosinophilic esophagitis, chronic inflammatory bowel diseases, and portal hypertension. The work is an attempt to integrate various recent and advanced AI techniques in health information systems.

Please consider the following comments. Thank you.

1. The Abstract does not include results. I recommend re-writing the abstract so that it refers to the key results/findings of this work (quantitative/qualitative results specific to this research).

2. The manuscript talks (in many sections) about “algorithms” in general; however, the manuscript does not present (in detail) any specific algorithm used in this work. This makes the text look like a textbook rather than a research paper. I recommend reducing the text talking about AI/algorithms in general and focusing on the research itself (i.e., presenting and commenting on specific algorithms used, strength/weakness of proposed algorithms and techniques, particular results obtained, examples of GDPR-compliant training algorithms, proving that the suggested functional architecture is successful, discussing the strengths and weaknesses of the proposed architecture, etc.).

3. No Related Work (or Literature Review) Section: Reviewing similar research in this area and comparing the proposed system/architecture with other architectures in the literature is essential to understand the contribution of this research.

4.  Line 99: The Synthetic Data Generator block ….

Explain more about how synthetic (artificial) data is generated and controlled to satisfy requirements.

5. Line 165: Missing data: This section talks about missing data in general, not the specific techniques used in this work (this is also related to comment 2 above).

6. Line 289: …Figure 3. Where is Figure 3? (I think you mean Figure 2).

7. Correction of some typos/errors:

- Line 24: legal and ethical implications arising form the sue of intelligent systems in clinical practice

- Line 32: Furthermore, even if the patient gives its consent, its data needs to be anonymised before being used to train ANNs… (an example correction: ...patients give their consent, their data ….).

- Line 292: the population identiied for the problem under consideration

Line 378: Portal hypertension (IP) is a pathological …. (is it IP?)

Moderate editing of the English language is required.

Round 2

Reviewer 1 Report

The authors addressed all my comments. Thank you.

Reviewer 2 Report

The authors have addressed the comments. Thank you.

 Minor editing of the English language is required.